# Solar radiation model and optimization of asymmetric large-span externally insulated plastic greenhouses

Chuanqing Wang[1]*, Kai Liu[1], Hongyu Ma[2], Tianhua Li[3], Shaojie Wang[4], Dalong Zhang[1], Min Wei[1]*

1 College of Horticultural Science and Engineering, Shandong Agricultural University, Taishan District, Shandong, China, 2 College of Information Science and Engineering, Shandong Agricultural University, Taishan District, Shandong, China, 3 College of Mechanical and Electronic Engineering, Shandong Agricultural University, Taishan District, Shandong, China, 4 College of Water Conservancy and Civil Engineering, Shandong Agricultural University, Taishan District, Shandong, China

* minwei@sdau.edu.cn (MW); 2020010073@sdau.edu.cn (CW)

## Abstract

To improve the light environment of asymmetric large-span externally insulated plastic greenhouses, a solar radiation model that considers the projection path equations of the insulation quilts and validated the model was established. The model was employed to investigate the impact of different heights, spans, and north lighting projection lengths on the greenhouses' light environment. The results revealed that ground radiation interception, a key component of winter lighting, was most influenced by height, followed by span, and least influenced by the projection length of the north lighting roof. Additionally, ground radiation spatial uniformity was most affected by height, followed by the projection length of the north lighting roof, and least influenced by span. The optimization objectives for solar radiation were set to maximize solar radiation interception and minimize the coefficient of variation. The optimal structural parameters for the asymmetric large-span externally insulated plastic greenhouse were determined using the NSGA-II method and the entropy weight-TOPSIS method: the height of 6.97 m, and the projection length of north lighting roof is 7.44 m for a greenhouse with a span of 20.00 m. Compared to the initial greenhouse, the optimized design enhances both radiation interception performance and ensures uniform light distribution. These results offer valuable theoretical guidance for greenhouse construction.

**Data Availability Statement:** All relevant data are within the manuscript and its Supporting Information files.

## Introduction

In recent years, rising labor costs have increased the need for mechanizing facility crop production, leading to higher demands for improvements in facility structures [1]. The Chinese solar greenhouse (CSG) is the main facility type for vegetable cultivation in northern China during the low-temperature season, characterized by good insulation and low energy consumption. However, it also has shortcomings, such as a low land utilization rate and in-convenient mechanization operations [2,3]. Consequently, in recent years, asymmetric large-span

**Funding:** This research was funded by China Agriculture Research System (CARS-23 to MW) and Key Research and Development Program of Ningxia Hui Autonomous Region (2023BCF01042 to MW). The funders had a role in study design, data collection and analysis, decision to publish, or preparation of the manuscript.

**Competing interests:** The authors have declared that no competing interests exist.

externally insulated plastic greenhouses have been developed and promoted as a novel domestic facility type [4]. However, some large-span externally insulated plastic greenhouses tend to be built blindly, leading to difficulties in ensuring the structural safety and environmental performance of the constructed facilities [5].

Solar radiation plays a crucial role in influencing the light and heat environment inside greenhouses, which significantly impact crop growth and production [6]. The interception of solar radiation n greenhouses is influenced by factors such as location, orientation, structure, and plant population distribution [7] Among these, the greenhouse's structure plays a decisive role in solar radiation interception [8,9]. Mellalou et al. [10] compared the solar radiation interception of even-span, uneven-span, and elliptical greenhouses and found that the east-west-oriented uneven-span greenhouse with a roof inclination of 12° harvested the maximum solar radiation. Xu et al. [11]studied the effects of orientation and structure on solar radiation interception in CSG and found that the ridge height remarkably influences the solar energy capture; comprehensive optimization of orientation and structure can increase the solar radiation interception of the rear wall by 3.95% compared to optimizing a single factor.

The above results are derived from studies on conventional plastic greenhouses and Chinese solar greenhouses. The asymmetric large-span externally insulated plastic greenhouse integrates curved light-harvesting surfaces at the northern and southern ends and a protective enclosure, distinctly differing from traditional plastic greenhouses and solar greenhouses [4]. Additionally, the large-span externally insulated plastic greenhouse is influenced by the top external insulation cover, forming a shadow zone in the northern part of the greenhouse, significantly impacting the uniformity of greenhouse lighting [12,13]. While many studies focus on shading from photovoltaic panels and the impact of shading in Chinese solar greenhouses on indoor environments and crop growth, research on the shading areas created by greenhouse insulation quilts are rarely reported [14–18]. Researchers have determined the appropriate structural parameters of asymmetric insulation plastic with the restrictive condition that the crops in the last row on the north side could receive sunlight throughout the day [13]. However, this approach does not quantify the amount of solar radiation intercepted. In conclusion, constructing a solar radiation model relying on the projection of the insulation quilt applied to asymmetric large-span externally insulated plastic greenhouses is deemed crucial.

The assessment criteria for light environment include radiation interception and spatial uniformity [19]. Coordination of achieving maximum radiation interception with spatial uniformity has emerged as a scientific issue requiring resolution for the efficient regulation of the light environment in greenhouses. Therefore, the optimization of the greenhouse structure is a multi-objective problem. With the sophisticated of computer technology, many scholars have utilized multi-objective optimization algorithms to refine both the structure and the environment of the greenhouse. Karambasti et al. [20] employed a multi-objective genetic algorithm (MOGA) to optimize design parameters for the Iranian region. Jiang et al. [21] applied the entropy weight method to optimize and retrofit a solar greenhouse in the severe cold climate of China, transforming it into a net-zero energy solar greenhouse (NZESG). While NSGAII, a classical algorithm, is widely used in the optimization strategy of greenhouse environmental control, its application to optimize the structural parameters of greenhouses remains relatively limited [22,23].

To date, most researchers have concentrated on examining the effects of the structural parameters of solar greenhouses and plastic greenhouses on lighting performance. However, developing a solar radiation model specifically for asymmetrical large-span externally insulated plastic greenhouses aimed at quantitatively analyzing and optimizing their structure has yet to be undertaken.

In summary, this study focuses on an asymmetric large-span externally insulated plastic greenhouse located in Jinan, Shandong, and constructs a solar radiation model that incorporates the projection shade of the insulation quilt. Using this model, we systemically analyze the impact of structural parameters on the light environment of the greenhouses. Three optimization objectives have been selected—ground cumulative radiation interception, ground average radiation interception, and coefficient of variation. The NSGA-II and the entropy weight-TOPSIS method were employed to optimize the structural parameters of the greenhouse. The findings from this study provide valuable guidance for the construction of more efficient greenhouses.

## Materials and methods

### Description of the plastic greenhouse

The experimental greenhouse in this study is an asymmetric large-span externally insulated plastic greenhouse that has undergone rapid development in recent years. The greenhouse is located in Jinan Laiwu Anxin Agricultural Technology Co., Ltd., Shandong Province, China (36°14′ N, 117°32′E). This asymmetrical greenhouse incorporates southern and northern transparent lighting roofs with enclosure structures at both ends (Fig 1). The ridge of the roof acts as a separator between the north and south lighting roofs, with their horizontal projections along the span direction denoting the projected length of the south lighting roof and the north roof, respectively. The greenhouse is oriented 6° west of south, with a ridge height of 6.5 m, a span of 20 m, and a length of 48 m. The projections of the lighting roof on the north and south sides are 7 m and 13 m, respectively (13 m + 7 m). The greenhouse construction features an arched steel truss wrapped with a 0.15-mm-thick polyolefin (PO) film. At night, a 3 cm thick insulation quilt made of sprayed cotton is applied to the lighting roof based on outdoor temperature conditions. The southern lighting roof and both ends of the side walls are normally uncovered during the winter season daytime, while the northern lighting roof remains covered to reduce heat loss effectively.

### Model construction

**Calculation of the incident global radiation.**   The amount of solar radiation received by the lighting roofs comprises three parts: direct radiation, diffuse radiation, and reflected

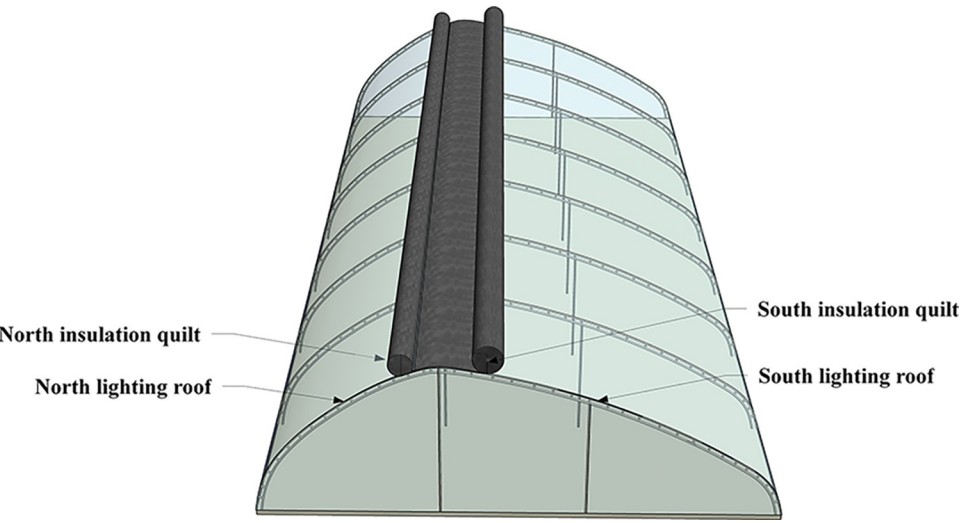

**Fig 1. Three-dimensional geometric model of large-span external insulation plastic greenhouse.**

radiation from the ground and objects on the ground. According to the method of Chen et al. [24], under the clear sky model, the direct solar radiation intensity $I_b$ at any point on the lighting roofs is expressed as:

$$I_b = I_{bn} \times cos\theta \tag{1}$$

Where $I_{bn}$ denotes the solar direct radiation intensity reaching an outside on typical clear days (W/m$^2$).

The angle of sunlight incidence on the lighting roof is closely associated with the solar altitude angle $h$, solar azimuth angle $A$ tilt angle of the lighting roof $\beta$, and greenhouse orientation $\gamma$. Eq (2) can be employed to calculate the sunlight incidence angle $\theta$.

$$cos\theta = cos\beta \times sinh + sin\beta \times cosh \times cos(A - \gamma) \tag{2}$$

Isotropic sky models are used for estimation of diffuse radiation[25]. Based on this simple isotropic sky model, the diffuse radiation and reflected radiation on the light roof can be obtained:

$$I_d = \frac{I_{dn} \times (1 + cos\,\beta)}{2} \tag{3}$$

$$I_r = \frac{(I_{bn} + I_{dn}) \times \rho \times (1 - cos\beta)}{2} \tag{4}$$

Where $I_d$ is the diffuse solar radiation reaching outdoors under typical clear sky conditions, $\rho$ is the outdoor ground reflectance, and 0.2 is taken in this study.

The outdoor solar direct radiation and diffuse radiation are calculated according to Bouguer's and Berlage's equations respectively [26].

$$I_{bn} = I_0 \times P^M \tag{5}$$

$$I_{dn} = \frac{I_0(1 - P^M)}{2 - 2.8\,lnP} sinh \tag{6}$$

$$I_0 = 1367 \times \left[1 + 0.033 \times cos\left(360 \times \frac{n-2}{365}\right)\right] \tag{7}$$

$$M = \frac{1}{sinh}(h > 30°), M = [1229 + (614\,sinh)^2]^{0.5} - 614\,sinh(h \leq 30°) \tag{8}$$

Where $I_0$ is the normal solar radiation in the upper atmosphere, which can be calculated by Eq (7). $P$ is the atmospheric transparency coefficient, in this study, take 0.73 [27]; $M$ is the mass of the atmosphere, corresponding to the distance of sunlight through the atmosphere, estimated by Eq (8).

Under cloudy weather conditions, the cloud-weather model can be supported by adjusting the solar radiation reaching the ground. The direct solar radiation $I'_b$ and diffuse solar radiation $I'_d$ can be obtained from the following equations, respectively:

$$I'_b = I_{bn} \cdot \left(1 - \frac{CC}{10}\right) \tag{9}$$

$$I'_d = CCF \cdot (I_{bn} + I_{dn}) - I'_b \tag{10}$$

 

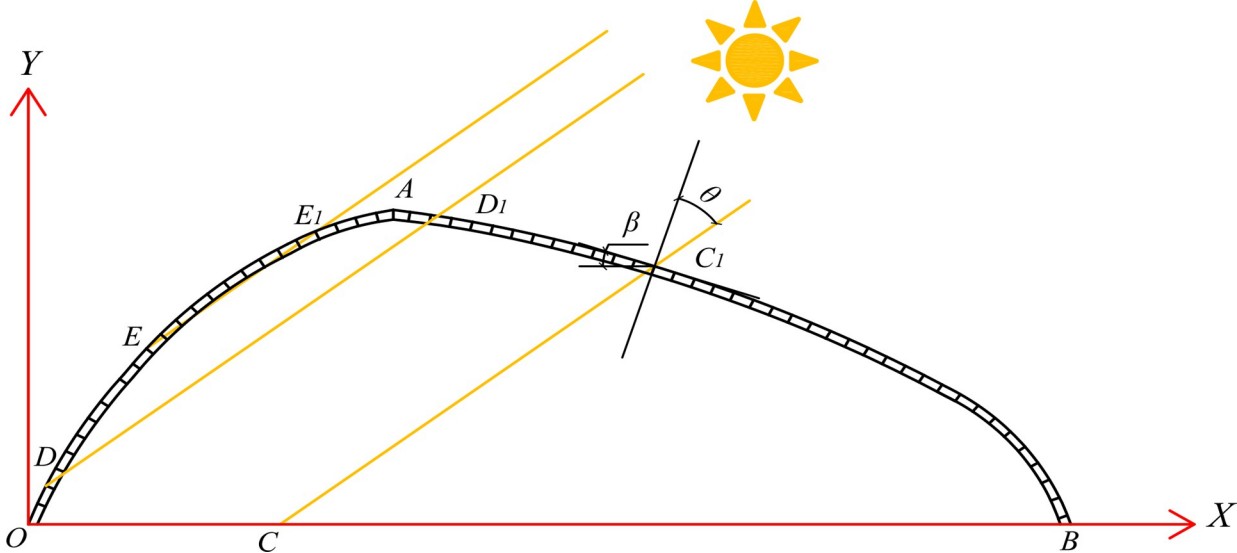

**Fig 2. Cross-section coordinate system of the greenhouse.**

In this equation, *CC* represents the cloud coverage ranging from 0 to 10, and *CCF* is the cloud cover coefficient. *CCF* is a function dependent on the cloud amount of *CC*, serving as the independent variable [28].

## Control equation of the transparent surface curve

The cross-section and rectangular coordinate system of the asymmetric large-span externally insulated plastic greenhouse is shown in Fig 2. The coordinate system is defined with the intersection point *O*, where the north lighting roof meets the ground. The X-axis represents the span direction, and the Y-axis represents the height direction. This study disregards the slight radian difference between the mathematical curve and the actual greenhouse skeleton. After fitting with the actual data, this study adopted the mathematical model of the parabolic lighting roof curve. The curve equations for the south and north lighting roofs can be obtained using the following formula:

$$Y_1 = a_1(x - x_A)^2 + y_A (x_A < x \le x_B) \tag{11}$$

$$Y_2 = a_2(x - x_A)^2 + y_A (x_O < x \le x_A) \tag{12}$$

$$a_1 = \frac{(y_B - y_A)}{(x_B - x_A)^2} \tag{13}$$

$$a_2 = \frac{(x_O - y_A)}{(x_O - x_A)^2} \tag{14}$$

Here, the coordinates of the origin *O* are represented as $(x_O, y_O)$. *A* represents the high point of the roof ridge, and *B* is the intersection of the southern lighting roof and the ground. The positions of *A* and *B* are variable and not fixed. The coordinates of points *A* and *B* are denoted as $(x_A, y_A)$ and $(x_B, y_B)$, respectively.

## Calculation of the internal global radiation

This study adopts the method of light reverse tracing of indoor coordinate points proposed in reference [29]. At a specific time $t$ during winter, a particular point $C$ on the ground can be matched with the corresponding incident point $C_1$ on the south lighting roof. Similarly, points $D$ and $E$ on the north lighting plane can be matched with incident points $D_1$ and $E_1$ on the south and north lighting roofs, respectively. Here, the coordinates of the corresponding incident points are represented as $(x, y)$, and the coordinates of arbitrary points on the ground and the north lighting roof are denoted as $(x_i, y_i)$, as shown in the following formula:

$$x_i = x_o + (y_i - y_o) \times \cos(\alpha - \gamma)/\tan h \tag{15}$$

$$y_i = y_o + (x_i - x_o) \times \tan h/\cos(A - \gamma) \tag{16}$$

The cover material on the surface of the greenhouse attenuates the global solar radiation passing through it into the interior of the greenhouse. The transmittance of greenhouse cover has a large influence on the total amount of global solar radiation in the greenhouse. The direct radiation transmittance $\tau_b$ of dry, clean cover material is related to the incident angle of sunlight, and its formula can be derived from Ma et al [29]:

$$\tau_b = \tau_{bn} \times (1 - \mu_1) \times (1 - \mu_2) \times (1 - \mu_3) \tag{17}$$

$\mu_1$, $\mu_2$, and $\mu_3$ are the shading loss of structural materials, transmission loss of covering material due to aging, and transmission loss of covering material due to dust and condensation, respectively; $\tau_{bn}$ is the transmittance of dry cleaning new covering material to direct solar radiation, which is calculated as follows

$$\tau_{bn} = \tau_{b0} \times \left[1 - 0.93^{(90-\theta)}\right] \times \left(1 - \frac{\theta}{1000}\right) \tag{18}$$

Where $\tau_{b0}$ is the transmittance of the dry cleaning new covering material to direct solar radiation when the incident angle is 0˚.

Similarly, the transmittance of diffuse radiation $\tau_{bn}$ when is equal to 0˚

$$\tau_{dn} = \tau_{d0} \times (1 - \mu_1) \times (1 - \mu_2) \times (1 - \mu_3) \tag{19}$$

The diffuse radiation transmittance $\tau_{dn}$ for the value of $\tau_{bn}$ when is equal to 0˚.

In a clear sky, the total radiation intensity of any point on the ground is equal to the sum of the direct and diffuse radiation intensity obtained by the point. In the same way, the total radiation intensity of any point on the north lighting roof is the sum of direct and diffuse radiation intensity on the north lighting roof.

$$I_{gb} = I_b \times \tau_b \times \sin h \tag{20}$$

$$I_{gd} = \frac{\lambda_g}{180} \times (I_d + I_r) \times \tau_d + I_b \times \tau_b \times \sin h \tag{21}$$

$$I_g = I_{gb} + I_{gd} \tag{22}$$

$$I_{nb} = I_b \times \tau_b \times \sin(\beta_n - h) \times \cos(A - \gamma) \tag{23}$$

$$I_{nd} = \frac{\lambda_n}{180} \times (I_d + I_r) \times \tau_d + I_b \times \tau_b \times \sin(\beta_n - h) \times \cos(A - \gamma) \tag{24}$$

$$I_b = I_{bn} \times cos\theta \tag{25}$$

Where $\lambda_g$, $\lambda_n$ represent the visible angles of the ground and north lighting roofs, respectively [30].

The daily cumulative intercept on the ground and the daily cumulative intercept on the north lighting roof are obtained by integrating Eqs (26) and (28). In addition, the average ground solar radiation interception and the average north lighting roof solar radiation interception are obtained according to Eqs (27) and (29).

$$G_g = \int_{t_1}^{t_2} \int_0^L I_g dLdt \times T \times 10^{-6} \tag{26}$$

$$G_{ga} = \int_{t_1}^{t_2} \int_0^L \frac{I_g}{L} dLdt \times T \times 10^{-6} \tag{27}$$

$$G_n = \int_{t_1}^{t_2} \int_0^{S_2} I_n dSdt \times T \times 10^{-6} \tag{28}$$

$$G_{na} = \int_{t_1}^{t_2} \int_0^{S_2} \frac{I_n}{S_2} dSdt \times T \times 10^{-6} \tag{29}$$

Where $T$ is every hour, 3600s; $t_1$, $t_2$ is the start and end time of the daylighting period of a greenhouse, which is the opening and closing time of the thermal insulation quilt in actual production [24]; $L$ is the span (m); $S_2$ is the arc length of the insulation for the north lighting roof (m), can be calculated according to the lighting surface integral formula.

## Insulation quilt projected path equation

The solar beams mainly transmit through the greenhouse lighting roofs. According to Fig 3, $Y_{O_1}$ and $Y_{R_1}$ are the equations for the circle center and tangent line through the southern lighting roof insulation, while $Y_{O_2}$ and $Y_{R_2}$ are the equations for the circle center and tangent line through the northern lighting roof insulation. The sun is essentially a point source of light, and the solar beams are considered parallel rays. Therefore, based on the formula for the distance between parallel lines, the equations for the tangent lines through the southern and northern insulation, $Y_{R_1}$ and $Y_{R_2}$, can be calculated. Additionally, $Y_{R_1}$ and $Y_{R_2}$ intersect with the $X$-axis at points $P_1$ and $P_2$, respectively.

$$Y_{R_1} = \frac{\tan h(x - L_2 - L_3) + (H - H_1 + R_1)\cos(A - \gamma)}{\cos(A - \gamma)} - R_1 \sqrt{\left[\frac{\tan h}{\cos(A - \gamma)}\right]^2 + 1} \tag{30}$$

$$Y_{R_2} = \frac{\tan h(x - L_2 + L_4) + (H - H_2 + R_2)\cos(A - \gamma)}{\cos(A - \gamma)} + R_2 \sqrt{\left[\frac{\tan h}{\cos(A - \gamma)}\right]^2 + 1} \tag{31}$$

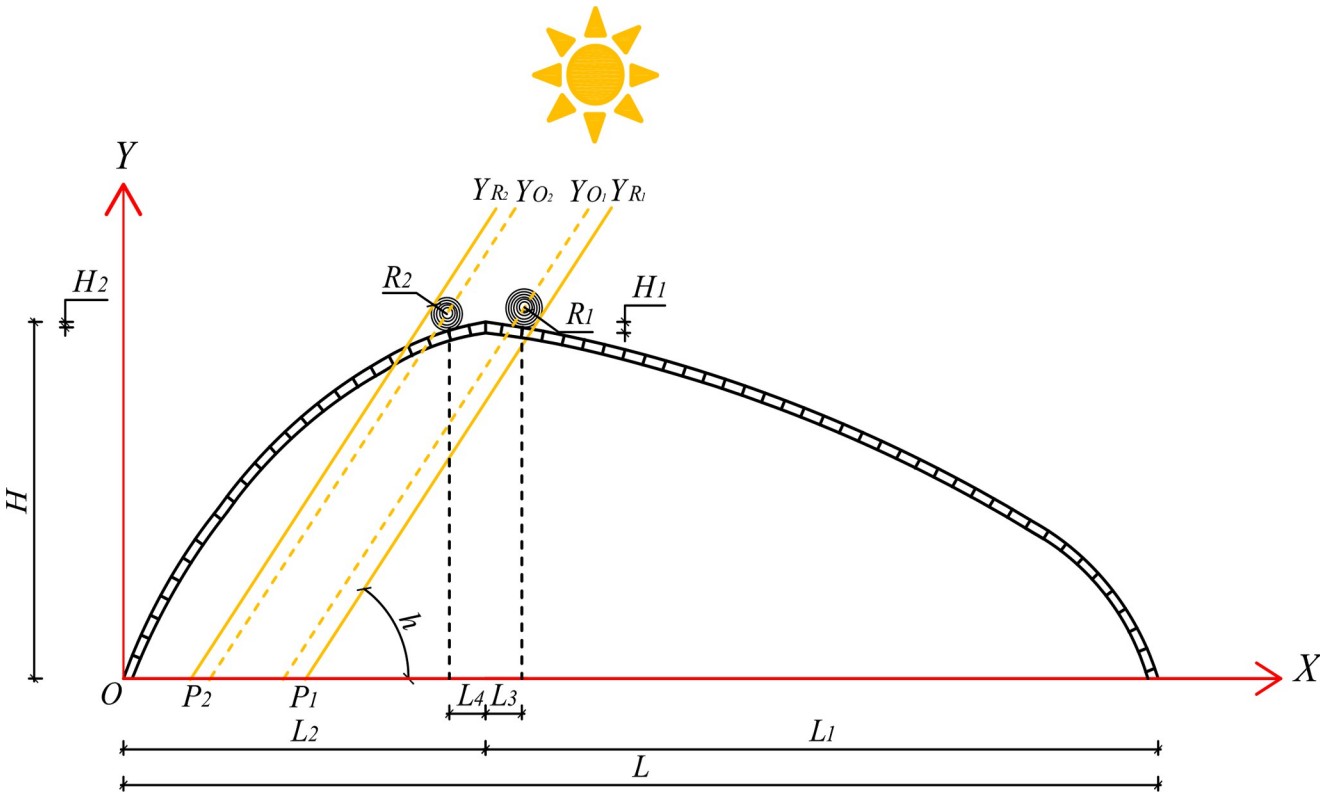

**Fig 3. Structure parameter diagram of greenhouse and solar beam projection diagram.**

$$P_1 = \frac{\cos(A - \gamma)\left\{R_1\sqrt{\left[\frac{\tan h}{\cos(A-\gamma)}\right]^2 + 1} - \frac{(H - H_1 + R_1)\cos(A-\gamma) - \tan h(L_2 + L_3)}{\cos(A-\gamma)}\right\}}{\tan h} \tag{32}$$

$$P_2 = \frac{-\cos(A - \gamma)\left\{R_2\sqrt{\left[\frac{\tan h}{\cos(A-\gamma)}\right]^2 + 1} + \frac{(H - H_2 + R_2)\cos(A-\gamma) - \tan h(L_2 - L_4)}{\cos(A-\gamma)}\right\}}{\tan h} \tag{33}$$

Where $H$ is the ridge height (m); $H_1$ the distance between the bottom of the south lighting roof insulation quilt and the ridge height (m), and $H_1$ takes the value of 0.15 m; $H_2$ is the distance between the bottom of the insulation quilt and the ridge height of the north lighting roof (m), and $H_2$ takes the value of 0.3 m; $R_1$ and $R_2$ are the radii of the insulation quilt of the south and north lighting roofs, respectively, and the insulation quilt is centered on the rolled-up axis according to the Archimedes-spiral (m) [31]; $L_2$ is the north lighting roof horizontal projection width (m); $L_3$ is the horizontal projection length of the bottom of the south insulation quilt from the ridge (m), $L_3$ takes the value of 0.7 m; $L_4$ is the horizontal projection length of the bottom of the north insulation quilt from the ridge (m), $L_4$ takes the value of 0.5 m.

The intersection points of $Y_{R_1}$ and $Y_{R_2}$ with the north lighting roof are $P_3$ and $P_4$ respectively. The analytical solutions for $P_3$ and $P_4$ can be obtained by solving the equations that combine the curve equation of the lighting roof and the projected path equation of the insulated quilt.

Influenced by the solar altitude angle, the projection of the insulation quilt will move either on the ground or the north lighting roof. Utilizing the path equation of the insulation quilt projection, the proportions of shaded area $S_g$ on the ground and $S_n$ on the north lighting roof were calculated, respectively:

$$S_g = \begin{cases} 0, P_2 < P_1 \leq 0 \\ P_1/L, P_2 < 0 < P_1 \\ (P_1 - P_2)/L, 0 < P_2 < P_1 \end{cases} \tag{34}$$

$$S_n = \begin{cases} 0, & P_4 < P_3 \leq 0 \\ \int_{P_3}^{P_4} \sqrt{1 + 2a_2(x - x_A)^2} dx/S_2, 0 < P_3 < P_4 \end{cases} \tag{35}$$

## Multi-objective optimization

The multi-objective optimization problem (MOP) involves optimizing multiple objective functions within a specified region. Various multi-objective optimization algorithms exist, and one of the most widely used intelligent optimization algorithms is the fast non-dominated sorting genetic algorithm with an elite strategy (NSGA-II) [32,33]. The algorithm initializes the population, obtaining the initial population through fast non-dominated sorting, selection, crossover, and mutation operations, with N individuals in the population. The parent and off-spring populations are merged, and the next generation of population individuals is derived through sorting and crowding calculations. This process continues until the maximum number of generations of evolution is reached and stopped [33]. In this study, the NSGA-II-based MATLAB multi-objective genetic algorithm tool, gamultiobj, was utilized to solve the optimization model. The parameters for Pareto score, population, and generation were set to 0.3, 120, and 200, respectively.

In this paper, we select the hypervolume (HV) to evaluate the performance of the multi-objective optimization algorithm. The hypervolume can evaluate the performance of convergence and diversity simultaneously [34,35]. The greater the hypervolume is, the better the convergence and diversity performance will be. The hypervolume can be defined as [36]:

$$HV(S) = VOL(\cup_{s \in S} [\lambda_1(s), Z_1^r] \times \cdots \times [\lambda_m(s), Z_m^r]) \tag{36}$$

where $VOL$ is the Lebesgue measure; $m$ is the number of objectives; $Z^r = (Z_1^r, \cdots Z_m^r)$ is a reference point in the target space.

## Objective functions

The ground and the north lighting roof constitute the primary components of the enclosure. Since the north lighting roof is made of film material, practical heat storage is not feasible at night. However, the ground is the sole source of heat storage and release, crucial for maintaining the thermal environment's stability in the greenhouse. Therefore, the evaluation index focuses on the ground light environment in multi-objective optimization.

The multi-objective optimization objective for large-span insulated plastic greenhouses aims to maximize the interception of ground solar radiation during winter while ensuring optimal uniformity within the greenhouse. Ground cumulative radiation and average

radiation maximization may be formulated as follows:

$$Maximize\, f_1 = G_g \tag{37}$$

$$Maximize\, f_2 = G_{ga} \tag{38}$$

The formula for coefficient of variation minimization is:

$$Minimize\, f_3 = \frac{\sqrt{\frac{1}{n-1}\sum_{i=1}^{n}(I_i - \bar{I}_g)^2}}{\bar{I}_g} \tag{39}$$

Where $I_i$ is the solar radiation intensity at the $i$th measurement point in the greenhouse (W/m$^2$); $\bar{I}_g$ is the average value of solar radiation in the greenhouse, (W/m$^2$). The coefficient of variation (C.V) is used to assess uniformity, and a higher coefficient of variation indicates poorer uniformity.

## Constraints

In severe winter conditions, asymmetrically insulated plastic greenhouses are equipped with insulation quilts outside the north lighting roof to reduce heat loss inside the greenhouse effectively. Previous studies have determined suitable structural parameters for asymmetric insulated plastic greenhouses, ensuring that the last row of crops on the north side receives sunlight at noon [13]. Simultaneously, this study considers the influence of azimuth angles, resulting in the horizontal projection length and ridge height of the north and south lighting roofs in asymmetrically insulated plastic greenhouses.

$$L_1 = \frac{Ltanhcos A - L_5 tanh + L_3 tanh + (H_1 + H_3)cos A}{tanh + tan\alpha cos A} \tag{40}$$

$$L_2 = \frac{Ltan\alpha cos A + L_5 tanh - L_3 tanh - (H_1 + H_3)cos A}{tanh + tan\alpha cos A} \tag{41}$$

$$H = \frac{[Ltan\alpha cos A - L_5 tanh + L_3 tanh + (H_1 + H_3)cos A]tan\alpha cos A}{tanh + tan\alpha cos A} \tag{42}$$

Among these parameters, $L_5$ represents the aisle width (m), with a specific value of 0.8 m. $\alpha$ denotes the reasonable south roof angle of the asymmetric externally insulated plastic greenhouse, determined based on the solar altitude angle and azimuth angle (˚) at 10:00 on the winter solstice day; $H_3$ represents the height of the crop, based on the seedling height of vegetables (m), with a given value of 0.2 m.

This study focused on asymmetric large-span externally insulated plastic greenhouses in Jinan City, Shandong Province, China. The research involved setting constraints on the structural parameters of these greenhouses based on different times for uncovering the thermal insulation.

Specifically, the winter solstice (12.21) was considered the period with the lowest solar altitude angle. During this period, the earliest uncovering of the insulation quilt occurs when the north lighting roof is not covered. Based on this time, the upper constraint of the projection length of the north lighting roof $L_{1max}$ and the lower constraint of the height $H_{min}$ are determined:

$$L_1 \leq L_{1max} \tag{43}$$

**Table 1. Decision variable constraints.**

| Span /m | 16 | 17 | 18 | 19 | 20 |
|---|---|---|---|---|---|
| Ridge Height/m | 4.5~5.6 | 4.7~5.9 | 5.0~6.3 | 5.3~6.6 | 5.6~7.0 |
| North lighting roof projection length/m | 5.0~7.2 | 5.3~7.6 | 5.6~8.1 | 6.0~8.5 | 6.3~9.0 |

$$H \geq H_{\min} \tag{44}$$

Additionally, the awakening of insects day (3.5), representing a crucial period of temperature warming, is considered the latest uncovering period of thermal insulation. Based on this, the lower constraint $L_{1\min}$ on $L_1$ and the upper constraint $H_{\max}$ on $H$ are determined:

$$L_1 \geq L_{1\min} \tag{45}$$

$$H \leq H_{\max} \tag{46}$$

The current span range for asymmetrical large-span external insulation plastic greenhouses, as applied in practice, is primarily 16 to 20 meters [37]. Therefore, the constraints for ridge height and north lighting roof projection length are calculated as shown in Table 1.

## Entropy weight TOPSIS decision-making

The NSGA-II optimization yields an optimal solution set, representing the Pareto optimal frontier solution set. Firstly, the entropy weighting method allocates weights to the function values within the multi-objective optimal solution set. Subsequently, TOPSIS determines the optimal solution. The TOPSIS optimization process involves normalizing the partition matrix, evaluating entropy weights, and determining the distances between the positive ideal solutions ($D_i^+$) and the negative ideal solutions ($D_i^-$) in the Pareto-optimal set, and calculating the relative closeness ($C$). For detailed steps, please refer to reference [38].

## Results and discussion

**Validation of the simulation model.** The testing period spanned from March 1 to December 31, 2021, during which solar radiation was monitored using the NHZ01 total solar radiation logger developed by Wuhan Neng Hui Technology Co., Ltd. The logger exhibits a measuring range of 0~2000 W/m², an accuracy of ±2%, and a display precision of 1 W/m². Operating within the temperature range of -40°C to +70°C, observational data were recorded every 15 minutes and regularly downloaded for subsequent analysis. The solar radiation recorder was placed 5 meters away from the base of the south-facing lighting roof and installed at a height of 1.5 meters.

To evaluate the accuracy of the solar radiation model, we employ the envelope method to validate the measured total radiation against simulated values at observation points within the 13m+7m greenhouse (Fig 4). This comparison was performed during specific periods, namely the summer solstice, autumn equinox, and winter solstice in 2021. The cloud data utilized in the experiment were sourced from the China Meteorological Data Service Centre, with all selected periods representing typical sunny days characterized by cloudiness levels ranging from 0 to 2. The evaluation of the solar radiation model for a large-span externally insulated plastic greenhouse was carried out using metrics such as mean absolute error (MAE), root mean square error (RMSE), and the coefficient of determination ($R^2$).

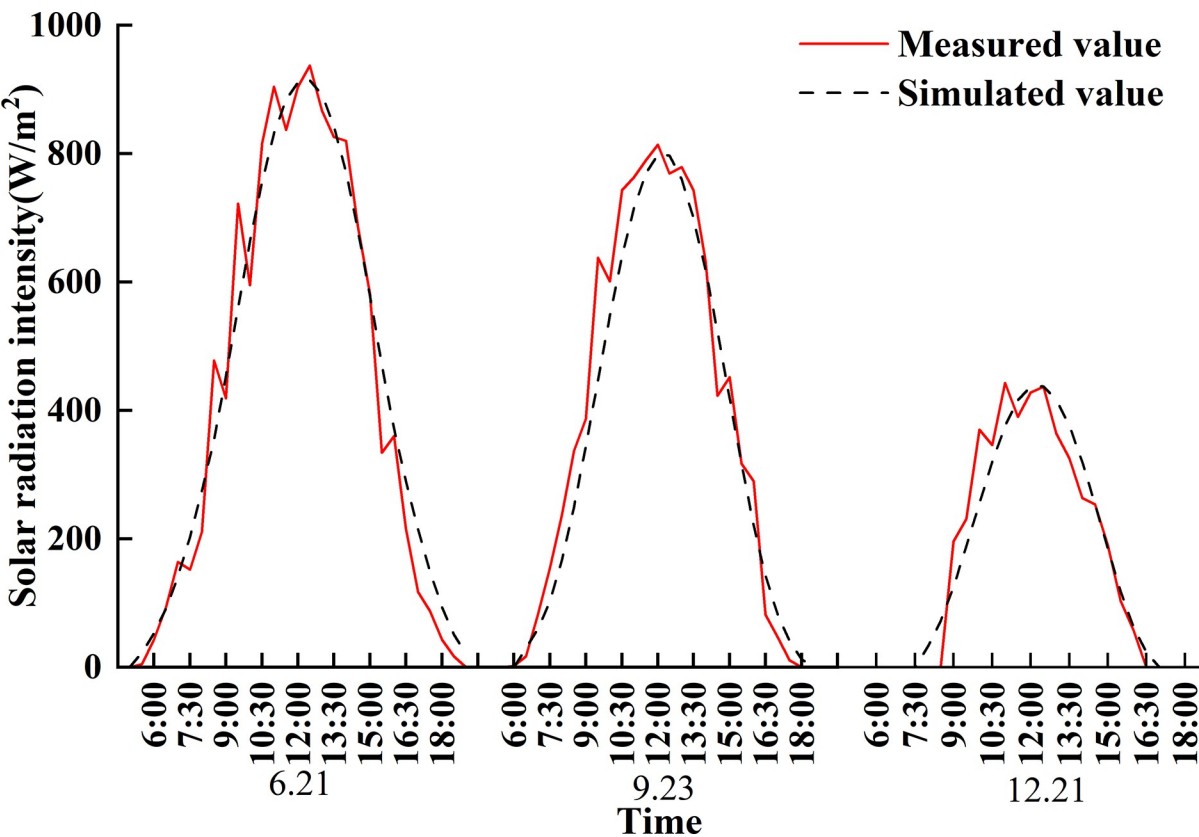

**Fig 4. Comparison of measured and simulated values for typical solar terms.**

The validation results indicate that the average absolute error (MAE) between the simulated and measured values of the 13m+7m greenhouse ranges from 37.2W/m² to 51.1W/m². In contrast, the root mean square error (RMSE) ranges from 47.9W/m² to 66.2W/m². The coefficient of determination ($R^2$) ranges from 0.90 to 0.96. In summary, the average MAE and RMSE values of the solar radiation model for large-span external insulation greenhouses are both less than 10%, and $R^2$ values remain above 0.9 and close to 1, indicating that the model is highly reliable.

**Shade validation.** To verify the accuracy of the projection path equations for insulation quilts, measurements of the projection profiles of greenhouse insulation quilts were conducted on clear days, specifically on March 22, 2024, and April 22, 2024. Fig 5 illustrates that the average absolute error (MAE) between these dates' calculated and measured values ranges from 0.10m to 0.12m. The root mean square error (RMSE) ranges from 0.11m to 0.12m. These results indicate the accuracy and reliability of the model's calculations.

Through comparison, it can be observed that on March 22, the shadow points gradually move northward as the solar altitude angle increases. This is because sunlight enters the greenhouse in the morning from the north lighting roof. Additionally, due to the greenhouse's orientation being 6° south of west, on April 22, the projection point of the insulation quilt is farther in the morning than in the afternoon.

The figure also indicates that the model's relative error at noon is significantly smaller than in the morning or afternoon. This is because there are clouds during both morning and afternoon, which weaken the intensity of solar radiation passing through the atmosphere. For example, on April 22, the maximum error at 16:00is 0.20 m.

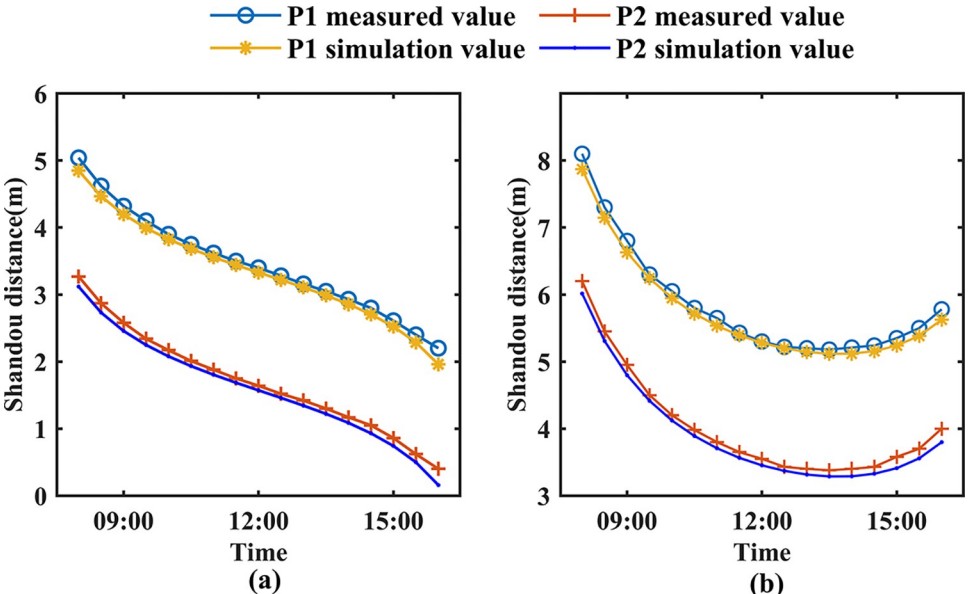

**Fig 5. Comparison of measured and simulated values of ground shadow contours.** (a) March 22, 2024; (b) April 22, 2024.

## Impact of structural parameters on light environment

The winter solstice, the shortest day of the year, is considered the time for evaluating light environments throughout the year. Hence, sunny conditions (CC = 0) during the winter solstice were chosen to assess light environment. Greenhouses with spans ranging from 16 to 20m, north lighting roof projection lengths between 1 and 9m, and ridge heights from 4 to 8m were selected as simulated structural parameters. This selection aims to investigate each parameter's impact on greenhouse light environment.

In winter, solar radiation primarily passes through the south lighting roof, reaching the ground and the north lighting roof. Analyzing the distribution of radiation interception on the roofs of different enclosure structures is crucial for optimizing the structural parameters of the greenhouse. Fig 6 illustrates the percentage of radiation interception on the south and north lighting roofs. The results reveal that ground radiation interception constitutes the predominant component, accounting for 75.3% to 95.0% of the total ground radiation interception and 5.0% to 24.7% of the radiation interception on the north lighting roof. This study also explored the impact of structural parameters on the light distribution on various surfaces. It was observed that the projected length of the north lighting roof had the most pronounced effect on light distribution. A smaller projected length of the north lighting roof led to a higher proportion of ground radiation interception; conversely, the radiation interception on the north lighting roof decreased. Increasing the height exhibited a decreasing trend in ground radiation interception, while the proportion of radiation interception on the north lighting roof increased. The span minimally influenced the ground and the north lighting roof, with the ground radiation interception ratio increasing with an increase in span. In contrast, the north lighting roof radiation interception ratio exhibited the opposite trend. As the ground surface serves as the sole source of heat storage and release in winter, the structural optimization design should maximize the ground radiation interception ratio to enhance the nighttime thermal insulation effect of the greenhouse [39].

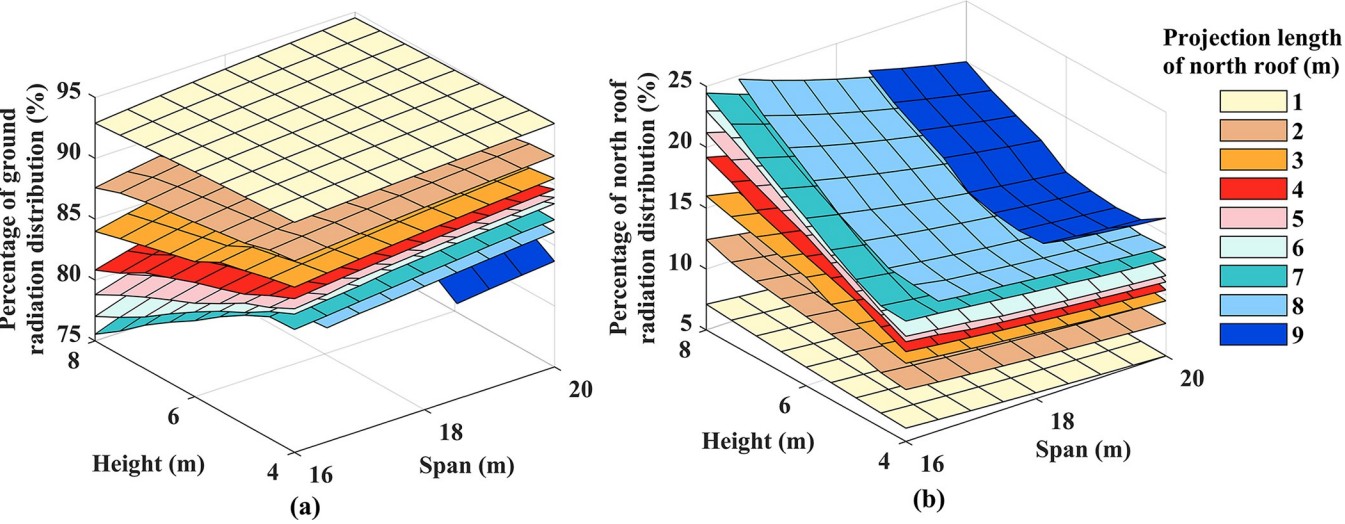

**Fig 6.** Solar radiation interception distribution on various surfaces of the greenhouse: (a) ground; (b) north lighting roof.

### Cumulative radiation interception

Fig 7 shows the impact of structural parameters on the cumulative radiation interception on the ground. It is worth noting that in a greenhouse with a height of 4 meters and a north lighting roof projection length of 9 meters, ground solar radiation interception is significantly reduced. Therefore, this study excludes the interference of these extreme parameters. The results indicate that the cumulative radiation on the ground exhibits a parabolic increase with height growth. For every 1 m increment in height, the range of variation in radiation interception increases from 3.22 MJ to 15.18 MJ, corresponding to a variation range of 2.56% to 12.71%. Conversely, the accumulated radiation on the ground shows a linear increase with an

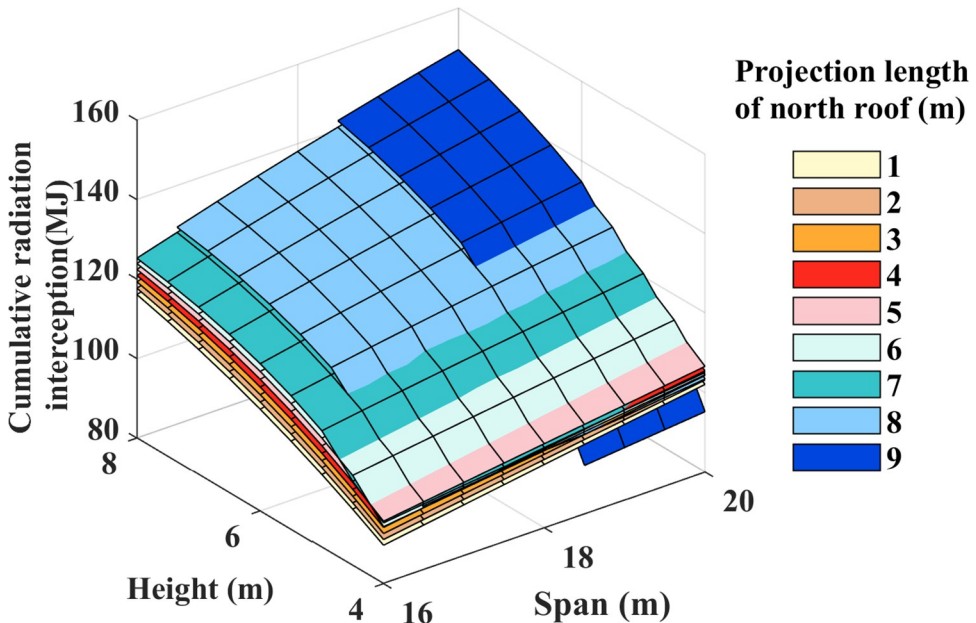

**Fig 7. Daily cumulative solar radiation interception at ground level.**

expanding span. With each 1 m increase in span, the variation range of radiation interception increase is 1.79MJ to 5.80MJ, corresponding to a variation range of 1.63% to 4.48%. Notably, the north lighting projection length change does not follow a consistent increasing or decreasing pattern. With each 1 m increase in the length of the north lighting roof projection, the change range of radiation interception increase is -2.25 MJ to 1.90 MJ, corresponding to a change range of -1.76% to 1.84%. In summary, ground radiation interception is most influenced by height, followed by span, and is least affected by the projection length of the north lighting roof.

During the winter solstice, when the north lighting roof projection length is between 6 to 9 meters, the interception of radiation by neighboring north lighting roof projections at heights of 4.3 m, 4.8 m, 5.3 m, and 5.8 m remains consistent across various greenhouse spans (Fig 7). A comparative examination of radiation spatial distribution was undertaken at noon on the winter solstice for greenhouses with varied construction to acquire a more profound knowledge of this phenomenon. Fig 8A and 8B show greenhouses with north-facing roofs that extend 8 meters (12m+8m) and 9 meters (11m+9m), each with a height of 5.3 meters. At noon, when the solar altitude angle is highest, the 12m+8m and 11m+9m greenhouses are shaded at the bottom edge of the north lighting roof and 1 meter away from it, respectively. The total radiation on the ground in the 11m+9m greenhouse is currently decreasing. Fig 8C and 8D illustrate greenhouses with a height of 5.8m, measuring 12m+8m and 11m+9m, respectively. It was observed that the shadows cast by their insulation covers precisely fell on the north lighting roof and the bottom foot of the north lighting roof. During this moment, the cumulative radiation on the ground of both greenhouses was equal. In summary, with an increase in the length of the north lighting projection, the shadow area gradually shifts southward, causing shade on the north side of the ground and a decrease in radiation interception. Conversely, increasing the ridge height causes the shadow area to shift northward, illuminating the north lighting roof and increasing ground radiation interception.

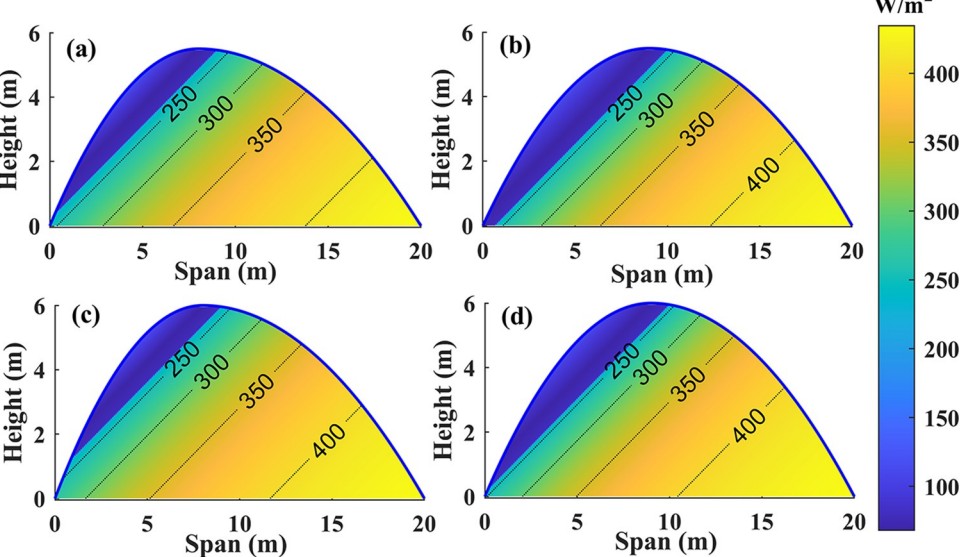

**Fig 8. Spatial distribution of different structured greenhouses at noon on the winter solstice (a): ridge height 5.3, 12m+8m; (b): ridge height 5.3, 11m+9m; (c): ridge height 5.8, 12m+8m; (d): Ridge height 5.8, 11m+9m.**

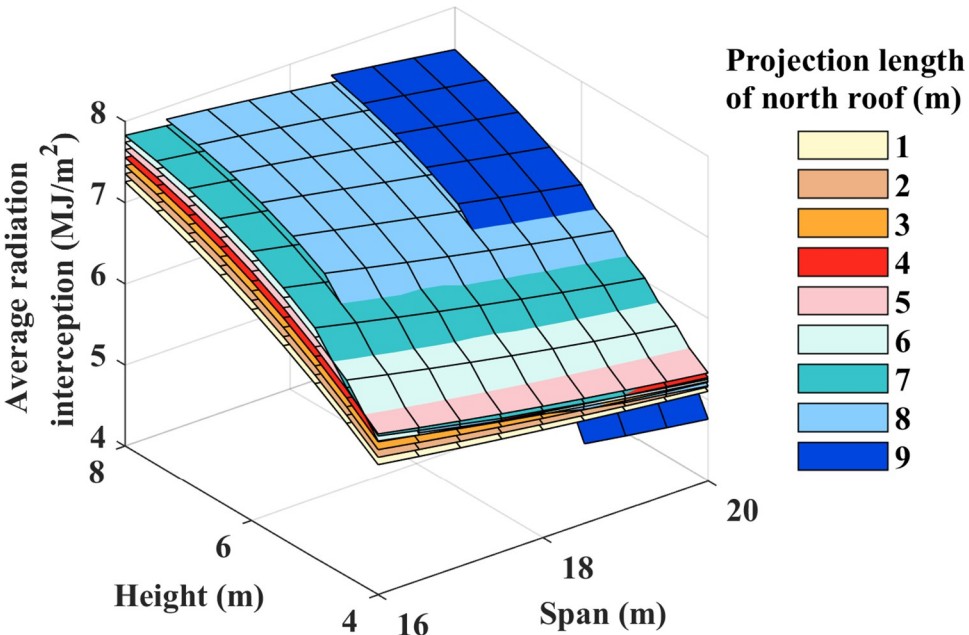

**Fig 9. Daily average solar radiation interception at ground level.**

## Average radiation interception

Comparing the actual radiation interception of different greenhouses proves challenging due to variations in their respective spans. Therefore, the average light interception index was employed to evaluate the greenhouse's radiation interception per unit area. Fig 9 provides the average radiation interception at ground level. As depicted in Fig 9, it is observed that the changes in height and the width of the north lighting roof projection follow the same trend as cumulative ground radiation interception, while the trend in span changes oppositely. This observation is consistent with the findings reported by Demir in the context of plastic greenhouses [40]. For every 1-meter height and span increase, the maximum average radiation interception increase is 12.71% and -3.49%, respectively. The variation range in average radiation interception for the north lighting roof projection aligns with cumulative radiation interception. In summary, it is evident that height has the most significant impact on average radiation interception.

## Spatial homogeneity

The assessment of the greenhouse light environment should encompass not only radiation interception but also the spatial homogeneity within the structure. In particular, the application of an outer insulation cover during winter proves advantageous in minimizing heat loss from within, even though it may lead to a reduction in light transmission on the northern side. When comparing the effects of different structural parameters on the coefficient of variation at the ground level (see Fig 10), it was observed that an increase in height resulted in a decrease in the coefficient of variation. With each 1-meter rise in height, the coefficient of variation decreased by a maximum of 12.02 percentage points. Conversely, the ground coefficient of variation displayed a linear increase with an expanding span, reaching a maximum increase of 1.26 percentage points for every 1-meter increment in span. The range of the coefficient of variation was -0.32 percentage points to 5.72 percentage points for every 1-meter increase in the

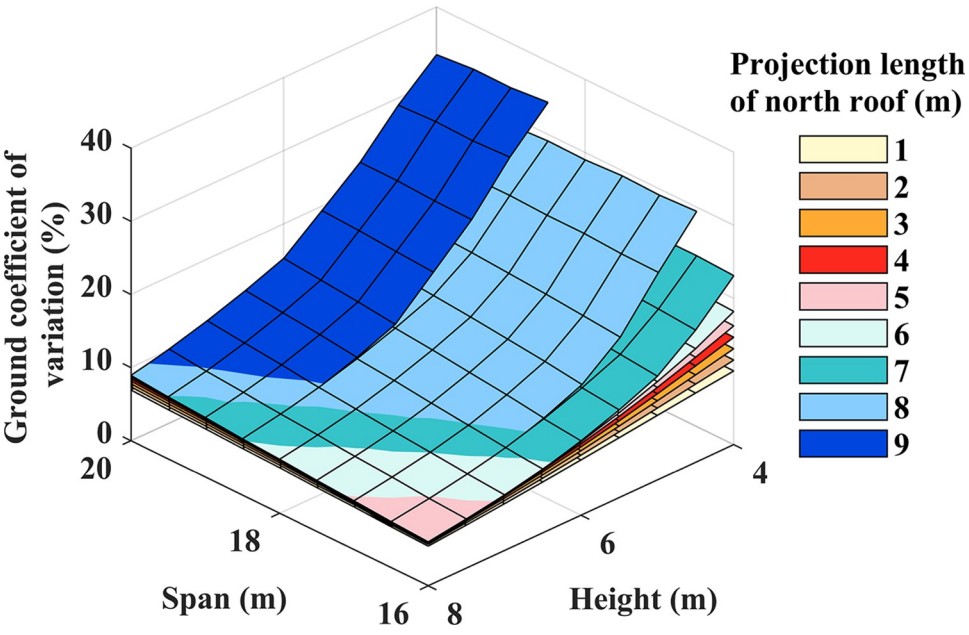

**Fig 10. Ground coefficient of variation.**

length of the north lighting roof projection. Clearly, raising the height effectively enhances soar radiation uniformity. However, an excessively high ridge height in the greenhouse may lead to a smaller insulation ratio, and increased heat dissipation, but may not be conducive to nighttime insulation [41]. Hence, a holistic evaluation is essential.

The significant variations in the coefficients of variation on the ground are primarily associated with the projected area of the insulation quilt. Examining the proportions of shadow areas on the ground and the north lighting roof at noon on the winter solstice, as depicted in Fig 11A and 11B, it becomes evident that when the length of the north lighting roof projection exceeds 5 meters, the ground is affected by the shadow of the insulation quilt, covering a

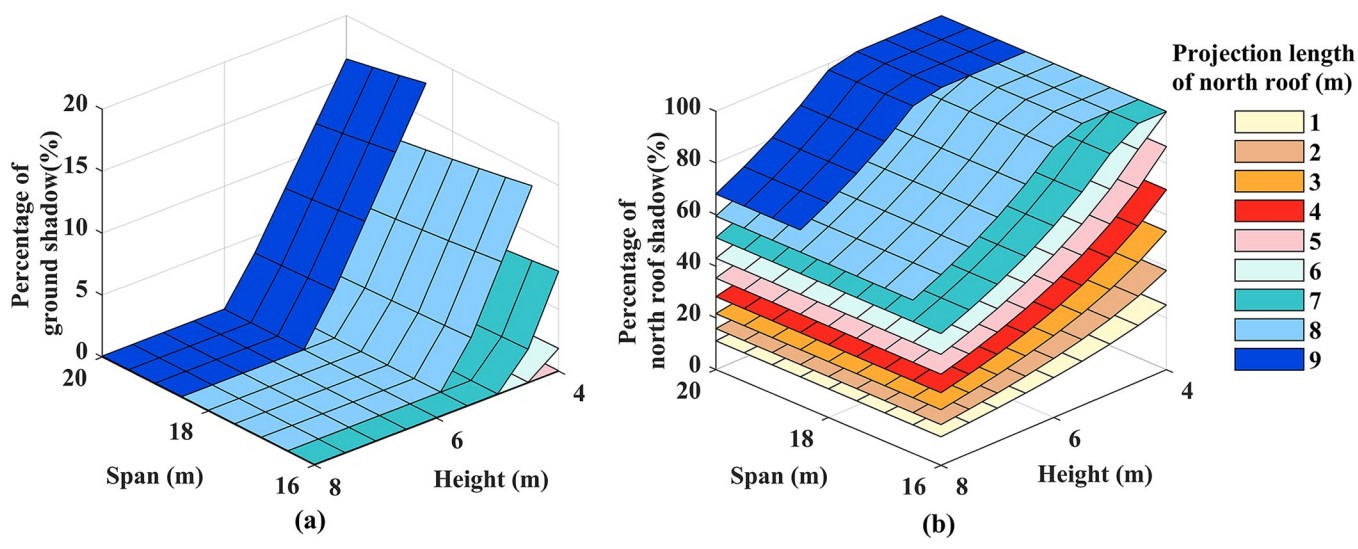

**Fig 11.** Percentage of shaded area for each surface:(a): ground; (b): north lighting roof.

maximum of 17.8% of the total cultivation area of the greenhouse. This shadow significantly impacts the growth and development of crops on the north side of the greenhouse [12]. As the height increases, the shadow of the insulation quilt is cast onto the north lighting roof. Consequently, there is a significant reduction in the proportion of the ground covered by the insulation shadow, Subsequently, the north lighting roof is lighted by solar beams, causing a quick decrease in the shadow area.

## Multi-objective optimization

**Convergence performance analysis.** The previous section analyzes the impact of structural parameters on greenhouse ground radiation interception and uniformity, identifying their interaction patterns. However, it falls short of presenting the optimal outcomes. Multi-objective optimization of asymmetric large-span externally insulated plastic greenhouses was performed using the NSGA-II algorithm, considering the defined objective functions and decision variables outlined in the "multi-objective optimization" section.

The figure shows the convergence curve of the objective function's hypervolume (HV) over 200 iterations for the NSGA-II algorithm. As illustrated in Fig 12, the convergence curve exhibits a rapid increase during the initial iterations, stabilizing after approximately 20 iterations. This indicates that the algorithm achieves convergence well before reaching the maximum number of iterations, demonstrating its efficiency and effectiveness in finding optimal solutions [35].

**Results of optimization.** The parallel coordinate plot in Fig 13 reveals that the span parameter in the solution set is clearly stratified due to the study setting the span as an integer. Simulation results indicate that the optimal heights of the greenhouse are all close to the upper limit of the constraints, suggesting that height is a key variable influencing daylighting

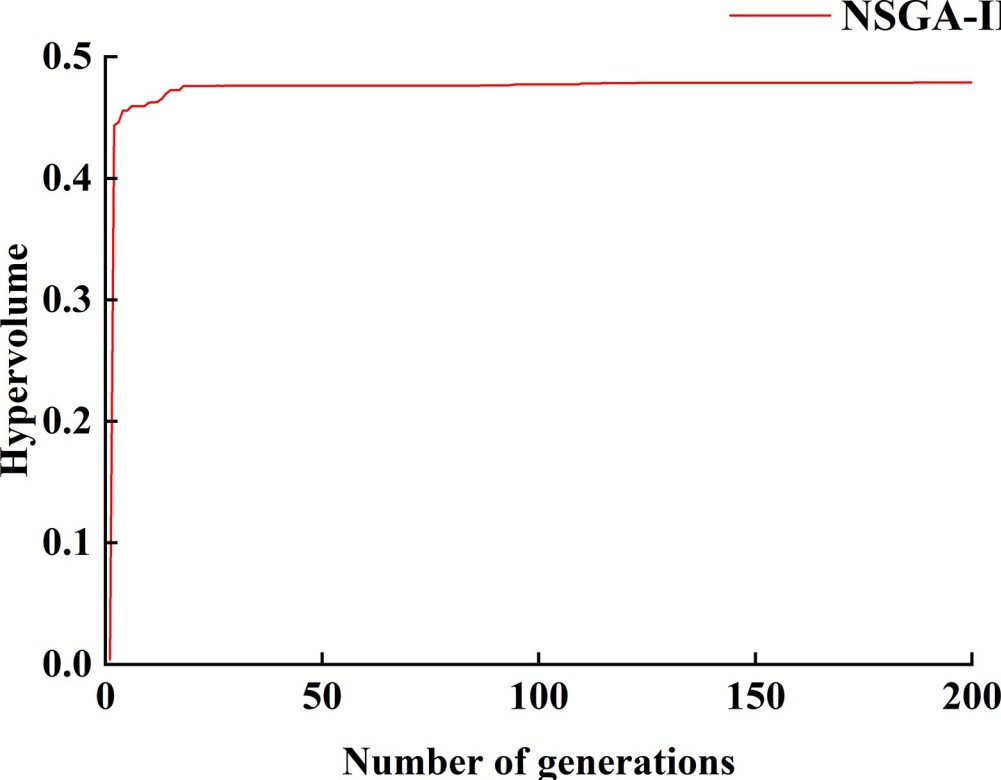

**Fig 12. Convergence curve of HV values of the NSGA-II algorithm.**

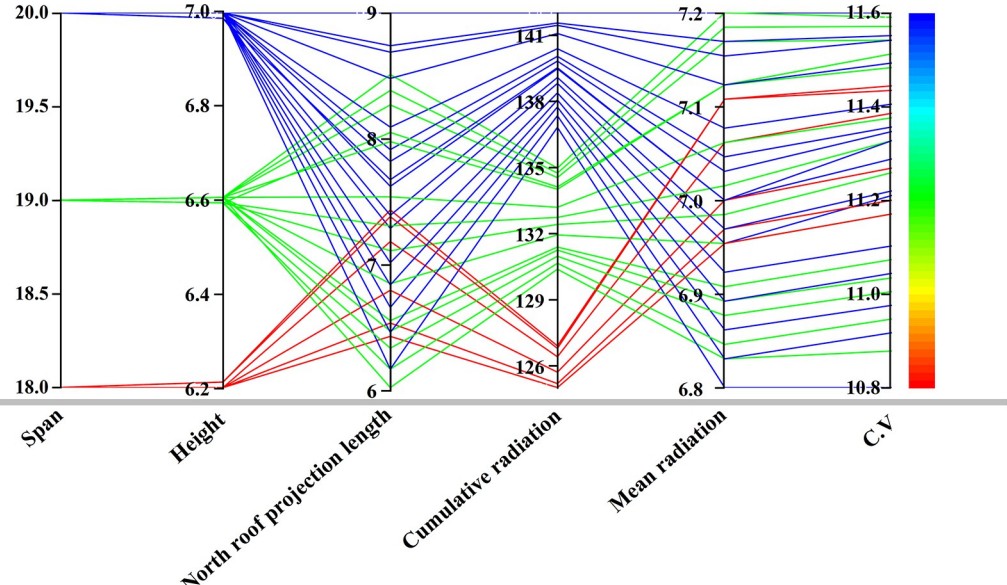

**Fig 13. Optimization results in a parallel coordinate plot.**

performance, consistent with the findings of Xu [11]. The plot also demonstrates that the solution set for the north lighting roof projection length is randomly distributed within the constraint range. On the other hand, there is a conflict between the cumulative radiation and average radiation in the objective functions. At the same time, there is a clear positive correlation between average radiation and coefficient of variation.

**Comparison of light strategies.** Fig 14 shows the 3D Pareto front obtained through computational analysis, comprising 36 decision solutions. Each point on the front represents a Pareto optimal solution distributed on the plane without inherent advantages or disadvantages. As shown in Fig 14, at design point A, the objective functions for cumulative radiation

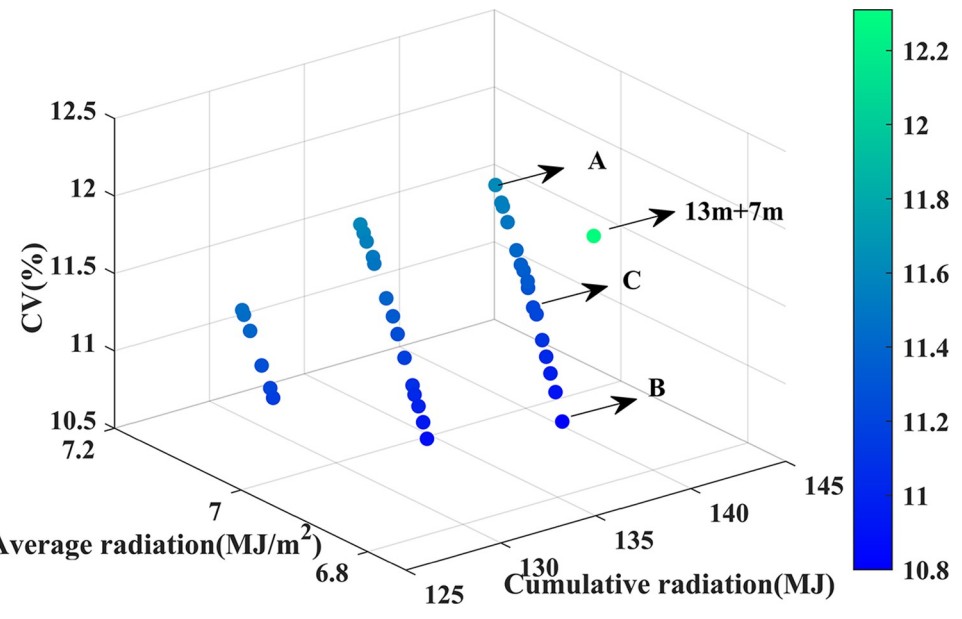

**Fig 14. Pareto-front 3D scatter plot.**

interception, average radiation interception, and the coefficient of variation reach their maximum values, which are 142.41 MJ, 7.12 MJ/m$^2$, and 11.62%, respectively. The corresponding decision variables are a span of 20m, a height of 6.97m, and a north lighting roof projection length of 8.98m. Conversely, at design point B, the average radiation interception and the coefficient of variation reach their minimum values, specifically 6.86 MJ and 10.80 J/m$^2$, respectively. At this point, the greenhouse structural parameters are a span of 20m, a height of 6.97m, and a north lighting roof projection length of 6.27m.

This study further analyzes the single-objective optimal solution. When maximizing solar radiation interception, the structural parameters approach their upper constraint limits, similar to the conclusion by Zhou [42] that an 11+9m greenhouse has the best heat storage capacity. The reason is that increasing the projection length of the north lighting roof while keeping the height constant reduces the light-transmitting area on the south side. Yet, the steeper roof angle improves solar radiation interception [43].

However, this study did not consider radiation uniformity. Due to the excessive width of the north lighting roof projection, the greenhouse's uniformity and effective solar radiation significantly decrease during the winter and spring seasons. The coefficient of variation is an important indicator for evaluating uniformity. At point C, uniformity is the best when the height is at its maximum, and the north lighting roof projection width is at its minimum. However, as shown in Fig 13, uniformity positively correlates with average radiation, meaning solar radiation interception decreases in winter.

In this article, the entropy weight TOPSIS is combined with the NSGA-II algorithm to choose the final optimum solution among available optimal solutions. First, the entropy weight method was used to assign weights to the daily cumulative solar radiation interception, average solar radiation interception, and the coefficient of variation objective functions, with weights of 29.9%, 32.6%, and 37.5%, respectively. Then, the Pareto optimal solutions were ranked according to these weights using TOPSIS. Finally, the optimal solution was identified at point C in Fig 14. Using the entropy weight TOPSIS method with NSGA-II, the optimal solution was determined to have a height of 6.97 m, a span of 20 m, and a north-facing lighting roof projection length of 7.44 m. For the multi-objective optimal solution, the ground cumulative solar radiation interception is 2.33% less than the maximum single cumulative radiation interception (142.41 MJ). However, the coefficient of variation is reduced by 0.40 percentage points. The coefficient of variation for the multi-objective optimal solution is 0.43 percentage points lower than the single maximum coefficient of variation (10.80%), while the ground cumulative radiation is increased by 1.67%.

Fig 14 also marks the original solar radiation interception and coefficient of variation for the 13m+7m greenhouse. Compared to the 13m+7m greenhouse, the Pareto-optimized solution significantly reduces the coefficient of variation and improves radiation interception. The comparative analysis between the optimization results and the initial greenhouse is shown in Table 2. In the multi-objective optimal solution, the ground cumulative radiation interception and average radiation interception for the 13m+7m greenhouse increased by 3.37% and 3.44%, respectively. At the same time, the coefficient of variation decreased by 1.13 percentage points. In summary, the multi-objective optimal solution not only improves ground radiation interception performance but also ensures excellent radiation uniformity, making it more suitable for practical production applications.

## Conclusion

In this study, a design methodology was developed to optimize the light environment of asymmetric large-span externally insulated plastic greenhouses. By calculating the projection path

**Table 2. Comparison analysis between optimization results and initial greenhouse.**

| | Factors | Initial greenhouse | NSGA-II coupled to entropy weight-TOPSIS |
|---|---|---|---|
| Decision variable | Span /m | 20.00 | 20.00 |
| | Ridge Height/m | 6.50 | 6.97 |
| | North lighting roof projection length/m | 7.00 | 7.44 |
| Objective function | Cumulative radiation (MJ) | 134.59 | 139.44 |
| | Average radiation (MJ/m$^2$) | 6.74 | 6.97 |
| | Coefficient of variation (%) | 12.31 | 11.23 |

equations for insulation quilt and employing light reverse tracing equations, a solar radiation model for asymmetric large-span externally insulated plastic greenhouses was constructed. The model was validated through practical measurements, achieving determination coefficients ranging from 0.90 to 0.96, indicating robust fitting performance. Utilizing this model, the study investigates the impact of different parameters (height, span, and north lighting roof projection length) on the light environment of asymmetric large-span externally insulated plastic greenhouses. Combining NSGA-II and entropy weight-TOPSIS multi-objective optimization methods, optimal solutions were obtained for radiation interception and spatial uniformity. The following conclusions were obtained.

1. During the winter solstice, ground radiation interception is the main source of winter solar energy, accounting for between 75.3% and 95.0%.

2. Height has the most significant impact on ground cumulative radiation interception and average radiation interception, followed by span, while the length of the north lighting roof projection has a minor effect. Similarly, the coefficient of variation is most influenced by changes in height, followed by the length of the north lighting roof projection, and least affected by span. The variations in radiation interception and the coefficient of variation were explained using projection path equations of the insulation quilts.

3. This study employed a multi-objective optimization approach to optimize greenhouses within the range of 16 to 20 meters. The results revealed that the optimal greenhouse configuration features a span of 20 meters, with an optimal height of 6.97 meters and a north lighting roof projection length of 7.44 meters. Compared to the initial greenhouse, cumulative radiation has increased by 3.37%, with a significant improvement in light distribution uniformity. These results offer valuable theoretical insights for the construction of greenhouses.

It is worth noting that this study focuses on constructing a solar radiation model for asymmetric large-span externally insulated plastic greenhouses in Jinan, Shandong, China, and proposes suitable structural parameters. Further research should explore the light and thermal environments in greenhouses across different geographical regions, as well as the optimization of structural parameters.

## Supporting information

**S1 Data.**
(ZIP)

## Author Contributions

**Conceptualization:** Hongyu Ma, Tianhua Li, Shaojie Wang, Min Wei.

**Data curation:** Chuanqing Wang, Kai Liu.

**Formal analysis:** Chuanqing Wang, Kai Liu.

**Funding acquisition:** Min Wei.

**Investigation:** Chuanqing Wang.

**Methodology:** Chuanqing Wang.

**Project administration:** Min Wei.

**Resources:** Min Wei.

**Software:** Chuanqing Wang, Hongyu Ma.

**Supervision:** Shaojie Wang, Min Wei.

**Validation:** Chuanqing Wang, Kai Liu.

**Visualization:** Chuanqing Wang.

**Writing – original draft:** Chuanqing Wang.

**Writing – review & editing:** Chuanqing Wang, Dalong Zhang.

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
