## [Decision Letter · Decision Letter 0]

12 Apr 2024

PONE-D-24-09089Solar radiation model and optimization of asymmetric large-span externally insulated plastic greenhousesPLOS ONE

Dear Dr. Wei,

Thank you for submitting your manuscript to PLOS ONE. After careful consideration, we feel that it has merit but does not fully meet PLOS ONE’s publication criteria as it currently stands. Therefore, we invite you to submit a revised version of the manuscript that addresses the points raised during the review process.

We look forward to receiving your revised manuscript.

Kind regards,

Ashfaque Ahmed Chowdhury, Ph.D., FHEA, FIEB

Academic Editor

PLOS ONE

Journal Requirements:

"This research was funded by China Agriculture Research System (CARS-23)"

5. We note that your Data Availability Statement is currently as follows: All relevant data are within the manuscript and its Supporting Information files.

7. PLOS requires an ORCID iD for the corresponding author in Editorial Manager on papers submitted after December 6th, 2016. Please ensure that you have an ORCID iD and that it is validated in Editorial Manager. To do this, go to ‘Update my Information’ (in the upper left-hand corner of the main menu), and click on the Fetch/Validate link next to the ORCID field. This will take you to the ORCID site and allow you to create a new iD or authenticate a pre-existing iD in Editorial Manager. Please see the following video for instructions on linking an ORCID iD to your Editorial Manager account: https://www.youtube.com/watch?v=_xcclfuvtxQ

8. We note that Figure 1 in your submission contain copyrighted images. All PLOS content is published under the Creative Commons Attribution License (CC BY 4.0), which means that the manuscript, images, and Supporting Information files will be freely available online, and any third party is permitted to access, download, copy, distribute, and use these materials in any way, even commercially, with proper attribution. For more information, see our copyright guidelines: http://journals.plos.org/plosone/s/licenses-and-copyright.

Reviewers' comments:

Reviewer's Responses to Questions

**Comments to the Author**

1. Is the manuscript technically sound, and do the data support the conclusions?

Reviewer #1: Partly

Reviewer #2: Yes

Reviewer #3: Yes

2. Has the statistical analysis been performed appropriately and rigorously? 

Reviewer #1: No

Reviewer #2: Yes

Reviewer #3: Yes

3. Have the authors made all data underlying the findings in their manuscript fully available?

Reviewer #1: No

Reviewer #2: No

Reviewer #3: No

4. Is the manuscript presented in an intelligible fashion and written in standard English?

Reviewer #1: Yes

Reviewer #2: No

Reviewer #3: Yes

5. Review Comments to the Author

Reviewer #1: I have reviewed this article about optimization of solar radiation in plastic greenhouses in a location in China.

I consider that the manuscript is interesting in its scope and objectives as it provides with valuable insights for development of agricultural facilities.

Regarding the English language, I do not see any particular flaws in the article.

That said, I consider that the method of finding the energy gains inside the greenhouse is not orthodox to say the least. First it seems that it is based in calculations and not in averaged official weather data. Secondly if the greenhouse is near transparent and its roofs are similar to horizontal surfaces with a (deliberately) neglected curvature, it is strange to distinguish between south and north.

Then a not very detailed geometric method of sunbeams is used which apparently disregards the possibility of clouded sky and we are not informed of the respective probability of overcast sky versus clear sky.

The term daylighting here is also misleading since we are dealing with thermal radiation in this case and illumination or visual perception seems out of the question.

I understand that the authors have used a model of calculation that might be common in China but it is not diffused outside this country and therefore it is not objectively validated.

It is surprising however, the high level of coincidence between the calculated and measured data in a just few days of monitoring.

I would prefer to have all the matter checked against weather data of radiation unless they are not available.

Finally, the former input of radiation is introduced in a kind of black-box energy or entropy simulator which is not very well known, at least outside China.

The results seem perhaps adequate but we do not know for instance what will happen with other greenhouse configurations or in different geographical areas.

Therefore, the tools presented and discussed are not entirely satisfactory in my humble opinion.

For the rest of the article the improvements and adaptions suggested to this greenhouse typology are minimal and do not entail a significant design process in which real advances for the construction and detailing of greenhouses are produced or enforced

Reviewer #2: This paper use the NSGA-II to maximizing solar raidation interception and minimizing the coefficient of variation. The application of this work is very interesting and this problem come from the real world engineering design application. However, this paper is very hard to read. The authors must be rewrite in some part before publication. My comments are below:

1) I reccommend the author to write the objective function in the standard mathematical optimization form. Is this work has 3 objective function? (Eq. 17, Eq. 19 and Eq. 21) I think the author must rewrite it in standard mathematic optimization form. It very hard to understand what is Eq. 17 19 and 27 and which objective are minimization or maximization.

2) The design lower limit and upper limit of each design variable must be shown in the mathematical form.

3) The convergence metric such as hypervolume must be shown to show the generation of the optimization method is enough.

4) The Parallel coordinate plot of the Pareto solution must be shown to show the relationship between the input parameter and output parameter of the Pareto solutions.

5) Table 2 is very hard to understand. If the author want the show the results of the single-objective optimization compared to the multi-objective optimization, I reccomend the author added the results of the single-objective optimization to the Pareto solution plot. Then the reader can understand the different.

6) This problem has 3 objectives, how or which criteria that the authors selected the design point from the Pareto solution?

Reviewer #3: Review Report of PONE-D-24-09089

a) The novelty of the greenhouse design compared with existing designs may be highlighted.

b) The research gaps may be mentioned pointwise at the end of Introduction.

c) Authors may state the sky model considered in the radiation modeling. The important equations may be referred in the text.

d) The model equations of the greenhouse may be referred in the text.

e) The convergence curve of the NSGA optimization may be added and discussed.

f) The major findings may be compared with published works.

g) The limitations and future scope may be added in Conclusions.

6. PLOS authors have the option to publish the peer review history of their article (what does this mean?). If published, this will include your full peer review and any attached files.

Reviewer #1: No

Reviewer #2: No

Reviewer #3: No

---

## [Author Response · Author response to Decision Letter 0]

24 May 2024

Dear reviewer,

Thank you very much for your hard work and patience. We have carefully revised the manuscript based on your comments, and the following is a point-by-point response to the comments.

For Reviewer #1:

1 Question: That said, I consider that the method of finding the energy gains inside the greenhouse is not orthodox to say the least. First it seems that it is based in calculations and not in averaged official weather data. Secondly if the greenhouse is near transparent and its roofs are similar to horizontal surfaces with a (deliberately) neglected curvature, it is strange to distinguish between south and north.

Answer: Thank you for the advice. The main models for outdoor solar radiation include the HOTTEL model[1], the ASHRAE model[2], the Liu and Jordan model[3], the Erbs model[4], and the Bouguer and Berlage formulas[5]. Among these, the Liu and Jordan model, the ASHRAE model, and the Erbs model are widely used. However, these models are based on radiation data from the United States and thus have regional applicability. On the other hand, the Bouguer and Berlage formulas can calculate solar radiation on horizontal surfaces if the local atmospheric transparency is clearly defined. Given that the atmospheric transparency used in this study is based on the latest observational research, it is feasible to adopt the Bouguer and Berlageis formulas.

I sincerely apologize for any misunderstanding. This study does not intentionally overlook curvature. Some researchers believe that, under fixed structural parameters, the radiation interception variations among solar greenhouses with various arched frameworks are relatively small[6-8]. Additionally, the discrepancy between the parabolic fitting equation and the actual frame curve is small. Thus, we use the parabolic equation as the roof equation, dividing the roof into south and north lighting roofs based on the ridge.

2 Question: Then a not very detailed geometric method of sunbeams is used which apparently disregards the possibility of clouded sky and we are not informed of the respective probability of overcast sky versus clear sky.

Answer: I sincerely apologize for any lack of clarity in conveying the content of the article. The classification of weather conditions primarily uses atmospheric transparency, clearness index, and cloud cover[4, 9, 10]. This study employs a cloud cover model, please refer to lines 166-173. Furthermore, the cloud cover data used for model validation is obtained from the China Meteorological Data Service Center, which is more accurate than manual assessment.

3 Question: The term daylighting here is also misleading since we are dealing with thermal radiation in this case and illumination or visual perception seems out of the question.

Answer: We acknowledge that "daylighting" is inappropriate and have made corrections in the paper accordingly.

4 Question: I understand that the authors have used a model of calculation that might be common in China but it is not diffused outside this country and therefore it is not objectively validated.

Answer: The model used in this study is widely applied in simulations of Chinese solar greenhouses, as referenced by Ma et al[11]. Additionally, Xu[12], Tian[13], and Xu[14] have empirically validated the model, demonstrating a good fit.

5 Question: It is surprising however, the high level of coincidence between the calculated and measured data in a just few days of monitoring. I would prefer to have all the matter checked against weather data of radiation unless they are not available.

Answer: Thank you for your suggestion. We have uploaded the raw data. It should be noted that the empirical data was recorded every 15 minutes. This study uses the envelope method to validate the measured total radiation at observation points in the 13m+7m greenhouse against simulated values at 30-minute intervals. As a result, we can observe a high degree of agreement between the calculated and measured data. Additionally, the study includes observations from March to December 2021, selecting the solstices and equinox (summer solstice, autumn equinox, and winter solstice) for validation, which provides a more representative analysis.

Prior to this study, the influence of azimuth angle on the projected path equation for thermal insulation was not taken into account. After recalculating, the formula can be found in lines 246-271. To further demonstrate the accuracy of the insulation projection equation, empirical data on shadow contours were supplemented on March 22nd and April 22nd. As elaborated in lines 400-417 the experimental fitting yielded satisfactory results.

6 Question: Finally, the former input of radiation is introduced in a kind of black-box energy or entropy simulator which is not very well known, at least outside China.

Answer: We utilized an empirical formula for radiation transmittance, which is suitable for various transparent covering materials. Previous researchers have employed computational methods, as indicated by[11, 15]. In the future, we plan to explore additional international methods for simulation.

7 Question: The results seem perhaps adequate but we do not know for instance what will happen with other greenhouse configurations or in different geographical areas.

Answer: This study focuses solely on structural optimization in the Jinan area of Shandong province. However, we have already begun analyzing the impact of orientation, shape, and position on greenhouses. Due to constraints in manuscript length, further exploration of these aspects is planned for future research.

Reviewer #2: 

1 Question: I recommend the author to write the objective function in the standard mathematical optimization form. Is this work has 3 objective function? (Eq. 17, Eq. 19 and Eq. 21) I think the author must rewrite it in standard mathematic optimization form. It very hard to understand what is Eq. 17 19 and 27 and which objective are minimization or maximization.

Answer: Thank you for your suggestion. We've selected cumulative radiation interception, average radiation interception, and coefficient of variation as our objective functions. Following your advice, we've restructured these functions into standard mathematical optimization forms, as elaborated in lines 311-322. 

2 Question: The design lower limit and upper limit of each design variable must be shown in the mathematical form.

According to your nice suggestions, I've made the necessary adjustments as per your guidance. Please kindly refer to lines 344-356 for the details.

3 Question: The convergence metric such as hypervolume must be shown to show the generation of the optimization method is enough.

Answer: We have already utilized the hypervolume indicator to evaluate the algorithm's performance. For further details, please refer to lines 539-551.

4 Question: The Parallel coordinate plot of the Pareto solution must be shown to show the relationship between the input parameter and output parameter of the Pareto solutions.

Answer: Following your suggestion, we have gone ahead and used a parallel coordinates plot to illustrate the Pareto solutions. Additionally, we have conducted an analysis of the relationship between these solutions' input and output parameters. You will find more information about this in lines 553-561.

5 Question: Table 2 is very hard to understand. If the author want the show the results of the single-objective optimization compared to the multi-objective optimization, I recommend the author added the results of the single-objective optimization to the Pareto solution plot. Then the reader can understand the different.

Answer: We've sincerely taken your advice to heart and made the necessary modifications to the images, along with providing a detailed analysis. For further insights, please refer to lines 564-604.

5 Question: This problem has 3 objectives, how or which criteria that the authors selected the design point from the Pareto solution?

Answer: We've employed the entropy-weighted TOPSIS method to select the optimal solution from the Pareto frontier. For specific details, please refer to lines 362-370. Additionally, insights into the analysis of the results can be found in lines 590-604.

 Reviewer #3: 

1 Question: The research gaps may be mentioned pointwise at the end of Introduction.

Answer: Thank you for your suggestion. This paper has outlined the gaps in previous research. For details, please refer to lines 100-104.

2 Question: Authors may state the sky model considered in the radiation modeling. The important equations may be referred in the text.

Answer: Following your helpful suggestions, this study has supplemented the isotropic sky model , Bouguer’s and Berlage’s equations. For details, please refer to lines 149-165.

3 Question: The model equations of the greenhouse may be referred in the text.

Answer: Following your suggestion, the reference formulas for the model equations have been supplemented. For details, please refer to lines 158-161,203-219.

4 Question: The convergence curve of the NSGA optimization may be added and discussed.

Answer: According to your nice suggestions, we have sincerely taken your advice and produced the HV convergence curves for the objective functions. The results can be found in lines 538-544.

5 Question: The major findings may be compared with published works.

Answer: According to your nice suggestions, we have incorporated the additional discussion content. For detailed information, please refer to lines 577-589.

6 Question: The limitations and future scope may be added in Conclusions.

Answer: Following your suggestion, this paper has added the limitations and future scope to the conclusion. For details, please refer to lines 645-651.

References

1. Hottel HC. A simple model for estimating the transmittance of direct solar radiation through clear atmospheres. Sol Energy. 1976;18(2):129-34. https://doi.org/DOI:10.1016/0038-092X(76)90045-1. 

2. American Society of Heating R, Air Conditioning Engineers I. ASHRAE handbook of fundamentals. Ashrae Handbook of Fundamentals. 1972. 

3. Liu BYH, Jordan RC. The Interrelationship and Characteristic Distribution of Direct, Diffuse and Total Solar Radiation. Sol Energy. 1960;4(3):1-19. https://doi.org/DOI:10.1016/0038-092X(60)90062-1. 

4. D. G, Erbs, and, S. A, Klein, and, et al. Estimation of the diffuse radiation fraction for hourly, daily and monthly-average global radiation. Sol Energy. 1982;28(4):293-302. https://doi.org/DOI:10.1016/0038-092X(82)90302-4. 

5. Handbook SEU. Japan Solar Energy Society. Ohmsha, Ltd. 1985:10-88. 

6. Xuan WY. Mathematical model establishment and analysis for greenhouse surface curve. Tianjin Agric Sci. 2006;12(4):3. https://doi.org/10.3969/j.issn.1006-6500.2006.04.016. 

7. Li JN, Ma CW, Zhao SM, Cui WH, X N. Light environment comparison of solar greenhouse with different roof shapes and inclination angles. Xinjiang Agricultural Sci. 2014;51(6):7. https://doi.org/10.6048/j.issn.1001-4330.2014.06.004. 

8. Tong GH, Christopher DM, Li TL, Wang TL. Passive solar energy utilization: A review of cross-section building parameter selection for Chinese solar greenhouses. Renew Sust Energ Rev. 2013;26:540-8. https://doi.org/10.1016/j.rser.2013.06.026. 

9. Duffie JA, Beckman WA, Blair N. Solar engineering of thermal processes, photovoltaics and wind: John Wiley & Sons; 2020.

10. Reindl DT, Beckman WA, Duffie JA. Diffuse fraction correlations. Sol Energy. 1990;45(1):1-7. https://doi.org/10.1016/0038-092X(90)90060-P. 

11. Ma CW, Zhao SM, Cheng JY, Wang N, Jiang YC, Wang SY, et al. On Establishing Light Environment Model in Chinese Solar Greenhouse. J Shenyang Agric Univ. 2013;44(05):513-7. https://doi.org/10.3969/j.issn.1000-1700.2013.05.001. 

12. Xu DM, Li YM, Zhang Y, Xu H, Li TL, Liu XA. Effects of orientation and structure on solar radiation interception in Chinese solar greenhouse. PLoS One. 2020;15(11):17. https://doi.org/10.1371/journal.pone.0242002. 

13. Tian D, Li Y, Zhao S, Wu Q, Ma C, Song W. An Analysis of the Influence of Construct Parameters on the Solar Radiation Input in an Insulated Plastic Greenhouse. Agronomy. 2024;14(3):510. https://doi.org/10.3390/agronomy14030510. 

14. Xu HJ, Cao YF, Li YR, Gao J, Jiang WJ, Zou ZR. Establishment and application of solar radiation model in solar greenhouse. Trans Chin Soc Agric Eng 2019;35(7):10. https://doi.org/10.11975/j.issn.1002-6819.2019.07.020. 

15. Yan QS, Zhao QZ. Thermal process of building. Beijing: China construction industry press.; 1986. 19-20 p.

We appreciate your consideration and constructive comments on our manuscript and we look forward to hearing from your positive answer. 

With our sincere respects. 

Dr. Pro. Min Wei

Dr. Chuanqing Wang

---

## [Decision Letter · Decision Letter 1]

12 Jul 2024

PONE-D-24-09089R1Solar radiation model and optimization of asymmetric large-span externally insulated plastic greenhousesPLOS ONE

Dear Dr. Wang,

Thank you for submitting your manuscript to PLOS ONE. After careful consideration, we feel that it has merit but does not fully meet PLOS ONE’s publication criteria as it currently stands. Therefore, we invite you to submit a revised version of the manuscript that addresses the points raised during the review process.

We look forward to receiving your revised manuscript.

Kind regards,

Ashfaque Ahmed Chowdhury, Ph.D., FHEA, FIEB

Academic Editor

PLOS ONE

Journal Requirements:

Additional Editor Comments:

Reviewer #1: The authors have satisfied most of my queries, still I do not see nephographs or false colour maps of the distribution of radiation in plan inside the greenhouse

Reviewer #2: I agreed with the author answer. Now, this paper is suitable to publish.

Reviewer #3: Authors have addressed my review comments successfully. However, the following comment was not addressed-

The novelty of the greenhouse design compared with existing designs may be highlighted.

---

## [Author Response · Author response to Decision Letter 1]

17 Jul 2024

Dear reviewer,

Thank you very much for your hard work and patience. We have carefully revised the manuscript based on your comments, and the following is a response to the comments.

Reviewer #1: The authors have satisfied most of my queries, still I do not see nephographs or false colour maps of the distribution of radiation in plan inside the greenhouse

Answer: Thank you for the advice. Critten's findings suggest that the length has little influence on the greenhouse's light transmission in the east-west direction [1]. Hence, this study treats the greenhouse length as infinite and focuses only on how its span affects the light environment. According to the solar radiation model, the spatial distribution of noon solar radiation for different structures of greenhouses on the winter solstice can be seen in Figure 8 (line 539).

Fig 8. Spatial distribution of different structured greenhouses at noon on the winter solstice.

(a): ridge height 5.3, 12m+8m; (b): ridge height 5.3, 11m+9m; (c): ridge height 5.8, 12m+8m; (d): Ridge height 5.8, 11m+9m.

 Reviewer #3: Authors have addressed my review comments successfully. However, the following comment was not addressed-The novelty of the greenhouse design compared with existing designs may be highlighted.

Answer: Following your helpful suggestions, the innovation of optimizing greenhouses compared with existing greenhouses is summarized as follows: This study proposes a new solar radiation model incorporating the projection path equations of the insulation quilts, combined with a multi-objective optimization method to optimize the structural parameters of large-span plastic greenhouses suitable for Jinan, China. The results indicate that the optimal greenhouse configuration features a span of 20 meters, a height of 6.97 meters, and a north-facing roof projection length of 7.44 meters. Compared to the initial greenhouse, cumulative radiation increased by 3.37%, and light distribution uniformity significantly improved. Please kindly refer to lines 605-615 and 641-647 for the details.

References

1. Critten DL. The effect of house length on the light transmissivity of single and multispan greenhouses. J Agric Eng Res. 1985;32(2):163-72. 

We appreciate your consideration and constructive comments on our manuscript and we look forward to hearing from your positive answer.

With our sincere respects. 

Dr. Pro. Min Wei

Dr. Chuanqing Wang

---

## [Editor Report · Decision Letter 2]

19 Aug 2024

Solar radiation model and optimization of asymmetric large-span externally insulated plastic greenhouses

PONE-D-24-09089R2

Dear Dr. Wang,

We’re pleased to inform you that your manuscript has been judged scientifically suitable for publication and will be formally accepted for publication once it meets all outstanding technical requirements.

Kind regards,

Ashfaque Ahmed Chowdhury, Ph.D., FHEA, FIEB

Academic Editor

PLOS ONE

---

## [Editor Report · Acceptance letter]

5 Nov 2024

PONE-D-24-09089R2 

PLOS ONE

Dear Dr. Wang, 

I'm pleased to inform you that your manuscript has been deemed suitable for publication in PLOS ONE. Congratulations! Your manuscript is now being handed over to our production team.

Kind regards, 

on behalf of

Dr. Ashfaque Ahmed Chowdhury 

Academic Editor

PLOS ONE